# Dairy Cows Activity under Heat Stress: A Case Study in Spain

**DOI:** 10.3390/ani11082305

**Published:** 2021-08-04

**Authors:** Adrián Ramón-Moragues, Patricia Carulla, Carlos Mínguez, Arantxa Villagrá, Fernando Estellés

**Affiliations:** 1Centro de Tecnología Animal CITA-IVIA, Polígono La Esperanza, 100, 12400 Segorbe, Castellón, Spain; ramon_adr@externos.gva.es (A.R.-M.); villagra_ara@gva.es (A.V.); 2Instituto de Ciencia y Tecnología Animal, Camino de Vera s/n, 46022 Valencia, Spain; pcarpas@doctor.upv.es; 3Departamento de Producción Animal y Salud Pública, Facultad de Veterinaria y Ciencias Experimentales, Universidad Católica de Valencia San Vicente Martir, Guillem de Castro 94, 46001 Valencia, Spain; cminguez.balaguer@ucv.es

**Keywords:** dairy cattle, behavior, circadian rhythms, heat stress, THI

## Abstract

**Simple Summary:**

Heat stress is a major problem for dairy cattle welfare, and it has several implications for milk production. In this work, automatic sensors were used to monitor cows’ activity. Different behaviors were recorded for 40 animals at different heat stress conditions. The main aim of the study was to use automatic sensors to identify behavior changes caused by heat stress on dairy cows. All behaviors studied were affected by environmental conditions. Animals adapted to heat stress by modifying their behavior, and automatic sensors provided valuable information in this regard. These findings might be the early development of an automatic early warning system for heat stress based on the behavioral modifications of dairy cattle.

**Abstract:**

Heat stress plays a role in livestock production in warm climates. Heat stress conditions impair animal welfare and compromise the productive and reproductive performance of dairy cattle. Under heat stress conditions, dairy cattle modify their behavior. Thus, the assessment of behavior alterations can be an indicator of environmental or physiological anomalies. Moreover, precision livestock farming allows for the individual and constant monitoring of animal behavior, arising as a tool to assess animal welfare. The purpose of this study was to evaluate the effect of heat stress on the behavior of dairy cows using activity sensors. The study was carried out in Tinajeros (Albacete, Spain) during the summer of 2020. Activity sensors were installed in 40 cows registering 6 different behaviors. Environmental conditions (temperature and humidity) were also monitored. Hourly data was calculated for both animal behavior and environmental conditions. Temperature and Heat Index (THI) was calculated for each hour. The accumulated THI during the previous 24 h period was determined for each hour, and the hours were statistically classified in quartiles according to the accumulated THI. Two groups were defined as Q4 for no stress and Q1 for heat stress. The results showed that animal behavior was altered under heat stress conditions. Increasing THI produces an increase in general activity, changes in feeding patterns and a decrease in rumination and resting behaviors, which is detrimental to animal welfare. Daily behavioral patterns were also affected. Under heat stress conditions, a reduction in resting behavior during the warmest hours and in rumination during the night was observed. In conclusion, heat stress affected all behaviors recorded as well as the daily patterns of the cows. Precision livestock farming sensors and the modelling of daily patterns were useful tools for monitoring animal behavior and detecting changes due to heat stress.

## 1. Introduction

The impact of hot weather conditions on livestock is increasingly important, mainly in light of climate change [1]. In addition to temperature, relative humidity plays a key role since it regulates the latent heat exchange from animals, thus conditioning their thermal balance. In this sense, the most widespread environmental indicator in regard to heat stress in animals is the Temperature and Humidity Index (THI) [2]. This index allows for the objective establishment of thresholds for heat stress conditions [3,4,5,6]. In the Mediterranean area, heat stress is a major threat to livestock farming [7]. In this area, the combination of high temperatures and high relative humidity can result in dramatic conditions for dairy cows [8]. In addition, variables related to the animal, such as the breed, productive level or physiological state, can affect the ability of animals to cope with heat, thus modifying the thresholds of THI in practice [9]. Thermal stress leads to production, reproduction and welfare problems [8,10,11], causing changes in the behavior patterns of animals [12].

Fully understanding the behavioral patterns of animals is essential, as variations or alterations of them can be an indicator of environmental or physiological anomalies [13,14]. In the case of dairy cattle, they spend about 3 to 5 h a day eating, between 7 and 10 h ruminating, 30 min drinking, 2 to 3 h being milked and require approximately 10 h of resting time [15]. Some studies have also investigated the daily distribution of these activities [16]. However, no mathematical models describing the daily distribution of these activities have been developed, nor has the effect of heat stress on these behavior patterns been studied. These models might help to identify variations in activity patterns as indicators of welfare or health issues induced by heat stress. Behavioral changes can occur when the environmental temperature exceeds the threshold held by the animal, thus resulting in heat stress, an early indicator of welfare, health and productive issues [10].

With a high THI, animals reduce rumination time, the volume of food eaten and the time they spend lying by altering their resting behavior and changing resting areas [11,17]. In this way, cows modify their feeding behavior, increasing water intake and change feeding times to cooler periods of the day [5,10]. In addition, activity patterns, the periods of minimum activity in the first hours of the day and maximum activity in the afternoon, are also modified [18].

The monitoring of animal behavior may help to identify problems or stressful situations for animals. Nevertheless, this is a tedious and time-consuming task when direct observation is considered, so this is where new technologies can play a significant role. These systems have made the modernization of livestock farms possible. Precision Livestock Farming (PLF) uses these advanced technologies to optimize the operation of farms through the individual monitoring of animals [19]. These types of systems have great potential to help farmers to raise animals in the best possible conditions [20] by supporting decision-making through the optimization of the information received [21,22]. In recent years, there has been an important progress in the development of sensors that allow us to monitor the activity, behavior, welfare, health and production of farm animals, as well as the surrounding environment. The more control the farmer has over the animals, the easier it is to detect unexpected behaviors and make decisions [23,24]. One of the species with the longest tradition of using this type of system is dairy cattle [25]. One of the key topics tackled by the PLF of dairy cattle is health improvement. Improving health conditions allows for the reduction of drug costs, the improvement of animal welfare, avoiding production losses, the increase in efficiency and, consequently, the improvement of the environmental, economic and social sustainability of dairy products [26]. In the market, we find some alternatives aimed at this purpose such as ear tags, necklaces or pedometers, among others [27]. These types of systems are based on the micro-movement patterns of the animals, which include behaviors such as eating, degree of activity, motor behaviors or even rumination. They establish a baseline of behavior at the individual level, and any deviation from normal behavior triggers an alert in the program [28].

The main aim of this work was to evaluate the effects of heat stress on the behavior of dairy cows using PLF technology. To this aim, two approaches were employed: on the one hand, the average time devoted to different activities under varying heat stress conditions was determined; on the other hand, mathematical models describing the daily patterns of different behaviors were developed for both sets of data (heat stress and no heat stress) and compared.

## 2. Materials and Methods

The study was carried out in a commercial dairy cattle farm (Friesian Holstein breed) in Tinajeros (Albacete, SE Spain) from 1 July 2020 to 18 September 2020 (summer conditions). The animals were housed in a free stall barn with cubicles. There were 7 independent pens, each of them with capacity for approximately 150 animals. The cows were milked three times per day (starting approximately at 7:00, 16:00 and 23:00). The feed (total mixed ration) was provided in feeding alleys after milking. The average (±sd) milk production per animal during this period was 38.27 ± 5.71 L/day, with an average (±sd) fat and protein content of 3.13 ± 0.67 g/L and 3.19 ± 0.49 g/L, respectively. The housing system was a common one for warm climates, consisting only of a covered space with no walls. Pens were separated by fences or feeding alleys. Two non-consecutive pens were monitored for environmental conditions and animal behavior. The pens had similar dimensions, each being 150 m long by 5 m wide, providing an average space of about 5 m^2^ per animal. Four automatic drinkers (2 m each) were placed in each pen. Fans were installed in the pens and operated automatically when the temperature was over 25 °C. Animals were cooled using misting fans in the waiting room before milking during the whole experimental period. The region where this livestock is found has an average temperature of 23 °C in the summer, with maximum temperatures of 33.2 °C and minimum temperatures of 13.8 °C. The average relative humidity at this time is 50%. During the summer (from July to September), an average precipitation of 54 mm accumulates.

To determine the environmental conditions, 15 temperature (T) and relative humidity (RH) sensors (Easy Log EL-UDB-2+, Lascar Electronics, UK) were distributed throughout the length of the pens at a height of 2.5 m (slightly above the animals to prevent them from interacting with the sensors). All sensors were programmed to gather information on T and RH every 5 min, and the hourly average values were calculated. The THI was calculated hourly as per the equation proposed by the National Research Council of the United States [29]:THI = (1.8 × T + 32) − (0.55 − 0.555 × RH) × (1.8 × T − 26)
where T is the dry bulb temperature (°C) and RH the relative humidity of the air (%).

The value of THI for a given hour may not reflect the actual heat stress suffered by an animal that is affected by the heat load during the previous hours [30]. In order to consider heat load, THI_load was calculated for each hour as the accumulated THI values were during the previous 24 h. To establish whether the data corresponded to a different level of heat stress, each hour of the days measured was classified into four groups according to the quartiles of THI_load. The value of the quartiles remains as follows: Q1: THI_load accumulated <1658.76; Q2: 1658.76 < THI_load accumulated < 1722.80; Q3: 1722.80 < THI_load accumulated < 1775.30; and Q4: THI_load accumulated >1775.30. The Q1 quartile corresponded to an average THI for the last 24 h of 69.12, while the Q4 quartile corresponded to an average THI of 73.97. Considering the THI thresholds established by St-Pierre et al. [30], both values rely on different heat stress risk categories and are thus considered in this work as different in terms of heat stress. Accordingly, the Q1 quartile was defined as no heat stress (NS) and the Q4 quartile as heat stress (HS).

Animal behavior was monitored using automatic sensors to record activity. At the beginning of the study in June 2020, 40 Tag cSense™ Flex Tag collars (SCR Engineers Ltd., Netanya, Israel) were installed in 40 animals, 20 at each pen. Considering the homogeneity of the animals within both pens, animals were randomly selected, resulting on average (±sd) in 2.81 ± 1.01 lactation (123 ± 23 days in milk). These sensors translate information on animal movements and micro-movements into different behaviors. The behaviors considered were eating, ruminating, resting, active behavior (any activity besides the described nutritional ones) and panting (heavy breathing). The sensors collect the information on activities every second and, as a summary, the system provides the number of minutes dedicated to each activity per animal at each hour. The behaviors are described in Table 1.

R software version 3.4.1 (R Core Team, Vienna, Austria) was used for all of the statistical analysis. A one-way ANOVA with a Fisher test was conducted to analyze the effects of heat stress on the time dedicated to each activity.

The statistical model used was:Yij = μ + Ti j + εijk
where:Yij = studied behavior (eating, rumination, rest, activity, and heavy breathing) in minutes for each hour (i) and animal (j);μ = general mean;Ti = heat stress level (NS/HS) for each hour (i);εij = residual error, residuals, deviations from the mean.

Subsequently, the behaviors were modeled in 24 h cycles confronting the accumulated THI values that corresponded to the Q1 and Q4 groups, looking for the possible effect of this on the different behaviors. To do this, non-linear regressions (using the software R project) [31] were used and the parameters for all models were obtained. Two different approaches were followed. When the behavior presented more than one peak during the day (Eating and Rumination), the approach followed by Villagrá et al. [32] was followed, by which the modelling equation was:Yij = μ + a × sin (2π/T × Hour + b) + c × cos (2π/T’ × Hour − d)
where:Yij = studied behavior (eating and rumination) in minutes for each hour (i) and animal (j);μ = daily average minutes devoted to the behavior;T and T’ = period length of the sin and cos waves;a,b,c,d = equation parameters;εij = residual error, residuals, deviations from the mean.

When a single peak was observed (Rest, Activity and Heavy Breathing), a simplified approach [33] was followed. The period of the model was set at 24 h in order to calculate the daily variation. The equation used was:Yij = μ + a × sin (2π/24 × Hour + b)
where:Yij = studied behavior (rest, activity, and heavy breathing) in minutes for each hour (i) and animal (j);μ = daily average minutes devoted to the behavior;a = modelled amplitude of daily variation;b = the hour at which the minimum value is achieved;εij = residual error, residuals, deviations from the mean.

In order to obtain the initiation parameters, the package nlswas used with the algorithm “brute force” [34]. In addition, the goodness of the model was evaluated according to the Coefficient of Determination ($R_{adj}^{2})$ that Spiess and Neumeyer [35] used, being:$$R_{adj}^{2}=\frac{1-\left(n-1\right)}{\left(n-p\right)\left(1-R^{2}\right)}$$
$$R^{2}=\;(1-\frac{RRS}{TSS})$$
where:*RSS* = Residual sum of squares;*TSS* = Total sum of squares;*n* = number of observations;*p* = number of parameters.

## 3. Results

### 3.1. Results of Time of Dedication Per Hour to Each Activity

Table 2 shows the mean values (±s.e.) of time (in minutes) per hour dedicated by the animals to each behavior.

Statistically significant differences were found for all behaviors recorded. As expected, the number of minutes with heavy breathing was higher for heat stress hours (HS) than for no heat stress (NS). This might be considered as indicating that the differences in heat stress between NS and HS were consistent. Heat stress resulted in less time devoted to eating and rumination, though the differences were small (about 0.5 min per hour). Higher differences were found for rest time, which was reduced by about 2.5 min per hour for HS. This result is in agreement with the differences observed for the aggregated activity (mid + high) time, which was found to be lower for NS.

### 3.2. Behavioral Daily Patterns

By adjusting the non-linear regression series to model the circadian rhythm of each of the behaviors, different equations were obtained. Table 3 shows these equations for each activity under both levels of THI. All behaviors were adjusted to periodic functions: rest behavior, added activity and heavy breathing were adjusted for periods of 24 h, while feeding and rumination behavior did not have a fixed period. This is because eating is influenced by the timing of the feeding of the animals.

With the equations previously discussed, graphs were made to observe the differences in each behavior between the group subjected to heat stress and the group without stress.

According to these models, eating behavior presents three peaks coinciding with the moment of the supply of the ration to the animals at the exit of the milking parlor. The ingestion time occurs earlier for HS than NS. In addition, the time dedicated to this behavior has lower minimums for HS than for NS (Figure 1a). In the case of rumination, we found two differentiated peaks throughout the day for both groups. The NS model presents a more pronounced peak in the early morning and a lower one in the late hours of the day, while the HS model shows a lower dedication to rumination during the night, whereas the afternoon peak starts earlier compared to NS (Figure 1b).

For rest behavior, it has been observed that, despite presenting similar patterns, the HS model represents lower rest time during the afternoon than the NS model (Figure 1c). This result is strongly related to activity behavior. The daily distribution of animal activity is similar for both groups, but the HS model shows higher activity values from the early morning to the early afternoon than the NS model (Figure 1d). Finally, in the case of heavy breathing, the NS model represents slight variation throughout the day. On the contrary, the HS model depicts a peak of heavy breathing from 12:00 to 20:00, coinciding with the hottest hours of the day (Figure 1e).

## 4. Discussion

Little data is available on the daily activity patterns for farm animals and the effects of heat stress. Changes in dairy cows’ behavior under heat stress conditions have already been reported in the literature [10]. Despite the THI equation used possibly presenting some constraints for the actual genetic strains of dairy cattle [36], the classification used in this work resulted in consistent differences between heat stress conditions. Heavy breathing might be considered a key indicator for heat stress, and the sensors used in this work have been satisfactorily validated as heat stress indicators through heavy breathing [37]. In this work, for the data categorized as HS, heavy breathing increased during the afternoon, coinciding with the warmer hours of the day. Lees et al. [38] observed similar results for dairy cows under heat stress conditions. Panting is an evaporative cooling process that is directly related to heat stress [5,39]. This cooling process is characterized by an increased respiratory rate with characteristic changes in breathing dynamics [12,40,41], by which body heat is released as the latent heat of the vaporization of moisture from the skin surface and the mucosa of the respiratory tract. This behavior is a fast and acute response to heat stress and thus occurs under high THI conditions, as depicted in this work.

In this study, it was observed that with a high THI, the animals reduced the time devoted to eating as well as the rumination time. This might be considered as a behavioral adaptation to heat stress since forage digestion leads to a large amount of metabolic heat production, which causes an increase in body temperature [42,43]. Thus, when environmental temperatures rise and reduce the ability of heat to dissipate, in order to keep the thermal balance, cows decrease their feed intake as a strategy to alleviate heat stress [17]. This is consistent with the results observed when attending to the daily eating patterns, where it can be observed that, under heat stress, animals tend to reduce their eating time during the afternoon, coinciding with higher THI values. This reduction is partially compensated during the night, when animals under heat stress conditions increase the time devoted to eating during the cooler hours of the day. The distribution of eating time for animals without heat stress followed a much more constant pattern. Three similar peaks corresponding with feeding time after milking were found, when a considerable amount of fresh feed is available. This is also consistent with the results obtained by Polsky et al. [5] and Fournel et al. [10] who found that, when the animals were exposed to heat stress, in addition to reducing the total feeding time, there was a shift in the feeding schedule towards cooler times of the day.

The negative correlation between rumination time and THI observed in this work has also been reported in the literature [7,44]. As observed in this work, this reduction in rumination time under heat stress conditions occurs during the night. During these cooler hours, when there is no heat stress conditions, animals tend to spend more time ruminating. During these hours, the highest peak in ruminating time was observed [45]. A secondary, much less intense peak is found during the afternoon. Under heat stress, animals maintain both rumination peaks, but the intensity of the one at night is much lower when compared to that under no heat stress conditions. This might be related to the heat production caused by rumination. Animals under heat stress conditions may reduce rumination during the cooler hours of the day in order to effectively reduce their metabolic temperature and compensate for daily heat stress.

Resting behavior is also affected by heat stress [46]. According to the literature, cows spend, on average, between 9 and 12 h per day resting [8]. That is an average between 22 and 30 min per hour. This behavior is an indicator of animal welfare in cattle, since when the animals suffer some type of stress or are not comfortable, it is strongly altered [47]. In this study, the resting time per daily hour was lower when heat stress occurred. This difference occurs mainly during the afternoon when animals under heat stress tend to rest much less than when no heat stress is a factor. As mentioned by Provolo and Riva [46] and De Palo et al. [48], cows under heat stress spend more time standing to achieve a greater heat dissipation through the skin, which is consistent with our study, the results of which indicate that resting time during the warmest hours is reduced under heat stress conditions. These results are consistent with the significant differences found for the activity level results. As previously described by Cook et al. [49] and Brzozowska et al. [50], animals increase their activity when THI increases. This lack of resting time is negative for cows, as it hinders the blood circulation in their udders [51], decreasing milk production [52] and increasing the risk of lameness [53].

In summary, the observed results show that the behaviors are related to each other such that feeding and activity increase, rumination and rest decrease and vice-versa. On the other hand, behaviors are influenced by environmental conditions. Increasing THI produces an increase in activity and changes in feeding patterns and a decrease in rumination and resting, impairing animal welfare.

## 5. Conclusions

Heat stress affected all behaviors recorded: heavy breathing, eating, ruminating, resting and activity. The higher the THI, the lower the time for feeding, rumination and resting. Panting and activity increase, as the animals remain standing for a longer time to dissipate heat. In addition, behavior patterns changed throughout the day depending on the heat stress suffered by the animals, occurring at cooler times of the day in the case of feeding, rumination and rest. PLF sensors and modelling daily patterns were useful tools for monitoring animal behavior and detecting changes due to heat stress.

## Figures and Tables

**Figure 1 animals-11-02305-f001:**
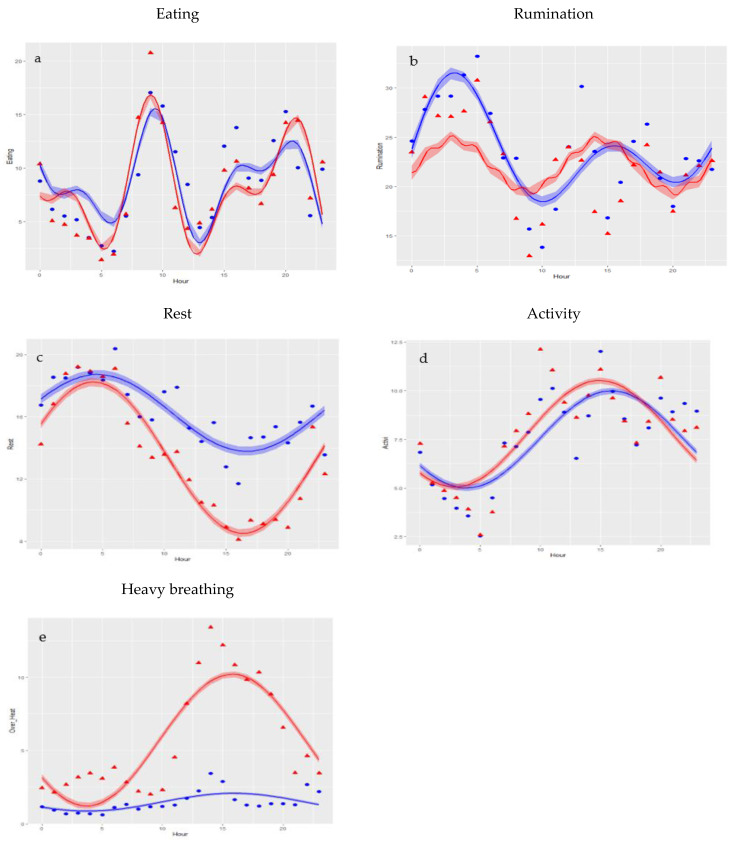
Modelled daily patterns for all behaviors considered under heat stress (red lines) and for no stress (blue lines) conditions. Shadowed areas represent modelling errors. (**a**) Eating, (**b**) Rumination, (**c**) Rest, (**d**) Activity, (**e**) Heavy breathing.

**Table 1 animals-11-02305-t001:** Description of the behaviors detected by the Tag cSense™ Flex Tag collars (SCR Engineers Ltd., Netanya, Israel).

Cow State	Definition
Mid activity	Combination of movements such as walking in an irregular rhythm or pattern or standing and performing various behaviors not characterized by intense and fast movements.
Rest	Detected when the animal is standing or lying down motionless and does not ruminate.
Rumination	Rhythmic circular movements of jaw not associated with eating, interrupted by brief pauses during the time that bolus is swallowed.
Heavy Breathing	High respiratory rate. Fast and shallow movement of the thorax visible when looking at the animal from the side, along with a forward heaving movement of body while breathing (standing or lying down).
High activity	Includes any combination of activities characterized by eruptive, intense and fast movements.
Eating	Muzzle or tongue physically contacts and manipulates feed, often but not always followed by visible chewing movements.

**Table 2 animals-11-02305-t002:** Average (±s.e.) number of minutes per hour devoted to each activity for animals under heat stress (HS) and without heat stress (NS).

Group	Eating	Rumination	Rest	Activity	Heavy Breathing
NS	8.74 ± 0.08	23.59 ± 0.11	16.16 ± 0.08	7.58 ± 0.05	1.50 ± 0.04
HS	8.27 ± 0.08	22.12 ± 0.11	13.36 ± 0.08	7.82 ± 0.05	5.74 ± 0.04
*p*-Value	<0.05	<0.05	<0.05	<0.05	<0.05

**Table 3 animals-11-02305-t003:** Equations modelling the daily patterns of each activity depending on heat stress conditions (NS = no stress, HS = heat stress).

Equations	R^2^
Eating_NS = 8.77 + 4.114 × sin (2π/9.82 × Hour + 2.052) + 2.814 × cos (2π/5.922 × Hour − 3.84)	0.74
Eating_HS = 8.142 + 3.710 × sin (2π/6.19 × Hour − 1.325) +5.006 × cos (2π/10.56 × Hour + 0.964)	0.80
Rumination_NS = 23.133 − 4.330 × sin (2π/18.628 × Hour − 97.534) − 5.150 × cos (2π/12.473 × Hour + 1.827)	0.75
Rumination_HS = 22.137 − 0.386 × sin (2π/1.808 × Hour − 12.284) − 2.660 × cos (2π/11.220 × Hour + 1.334)	0.76
Rest_NS = 16.247 − 2.466 × sin (2π/24 × Hour − 2.763)	0.73
Rest_HS = 13.371 − 4.872 × sin (2 π/24 × Hour − 2.680)	0.79
Activity_NS = 7.503 + 2.496 × sin (2π/24 × Hour − 2.568)	0.77
Activity_HS = 7.796 + 2.735 × sin (2π/24 × Hour − 2.313)	0.77
Heavy breathing_NS = 1.476 + 0.608 × sin (2π/24 × Hour − 2.565)	0.93
Heavy breathing_HS = 5.71 + 4.492 × sin (2π/24 × Hour − 2.535)	0.58

## Data Availability

Restrictions apply to the availability of these data. Data was obtained from SCR Engineers Ltd. (Netanya, Israel) and are available from the authors with the permission of SCR Engineers Ltd. (Netanya, Israel).

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
