# Peer review of "Dairy Cows Activity under Heat Stress: A Case Study in Spain"

_animals, 2021, doi:10.3390/ani11082305_

Round 1

Reviewer 1 Report

The present study aimed to develop mathematical models describing daily patterns of different behaviors of dairy cows and to evaluate the effect of heat stress on the behavior of dairy cattle using PLF sensors. 

I believe the research is interesting, but there are few points that should be addressed prior to acceptance.

  1. The abstract should be changed, as it refers to Temperature and Heat Index, and this is something else than THI;
  2. Please check your objective as it appears to analyze the behavior using the developed equations, and not the other way around;
  3. Some considerations should be addressed by the fact that THI was proposed in the 70s, and today's cows are not the same as back then. I believe this should be stated as a limitation of the study;
  4. Please consider nominating the figures as a,b, c, etc, and the analyzed behaviors should be in the figure heading;
  5. Please consider (either in the M & M session or the Results session) describing better the overall environmental data of the region, such as the mean daily temperature during summer, etc.;
  6.  It is not clear whether your data is the value +_ sd or se, as both are mentioned in the text. Please verify it.

Author Response

Authors' would like to thank the reviewer for the valuable comments received that have allowed to improve the quality of the document. Please find attached the document with detailed responses.

Reviewer 2 Report

Title

I'd suggest to use cows instead of cattle.

Materials and methods

Authors should provide relevant information about the farm involved in the experimental trial: cow breed, performance of the herd (average milk yield, average fat and protein content), management practices (feeding system, time of feeding distribution, formulation and proximate composition of the diet, number of milking during the day, milking system, etc.), equipments (drinking places and their measures, any ventilation and cooling systems, etc.).

Furthermore, the Authors should also provide data regarding the 40 selected animals (how and why they select those cows?), describing their characteristics: parity, stage of lactation, productive performance, etc. 

Regarding the data collected by the sensors, please indicate how often the sensors detected the behaviours (e.g., every x minutes).

When referring to behaviours as analysed variables, the Authors should always used the same words: Rumination or Ruminating, Resting or Rest, Heavy breathing or Panting).

Statistical models. The Authors stated that Eating and Rumination behaviours presented more than one peak during the day, but when describing the used equation they reported all the considered behaviours instead of just the two behaviours mentioned above. Same happened with behaviours showing a single peak.

Results and discussion

Heat stress is a major concern in dairy farming, and it will be even more in the future because of climate changes. Heat stress is relevant due to its negative impact on dairy cows, starting from the productive performance (i.e., milk yield and quality), with economic consequences for the farmers. What is missing in the Manuscript is explaining and supporting the behavioural results with some productive traits that for sure are available in the farm.

Finally, please check the English language, the Manuscript is rife with small errors and oversight: 

  • pag. 4: [...], modelling tge equation as [...]
  • pag. 4: When a single peak as observed [...]
  • pag 5:  unnecessary space before In order to get the initiation...
  • pag. 5: Table 2 caption repeated twice
  • pag. 5: Table 2. P-Valor...?
  • pag. 6: unnecessary space before 3.2 Behavioural daily patterns
  • pag. 8: In this study in was observed [...]
  • pag 8: Animals under hear stress conditions [...]

Author Response

(The authors gave the same response as above.)

Round 2

Reviewer 2 Report

The commercial farm in which the Authors carried out their experimental trial rears cows of high genetic merit: average MY at farm level ranges 32.6 to 44.0 kg/head*day in the summer period (Jul to Sep) when performance are expected to be (negatively) affected by environmental conditions. Therefore, it's plausible that the farmer implements several strategies to prevent heat stress to his/her animals: the Authors, in fact, added to the manuscript that fans are installed in the pens and cows are cooled with misting fans before entering the milking room. Without any objective confirmation (milk yield or other parameters –internal temperature– affected by HS) how the Authors can affirm (and support) that the detected conditions of temperature and humidity really caused heat stress to cows during the trial? Otherwise, how they can state with certainty that observed variations in behaviours are affected by THI rather than by other confounding factors?

The Authors correctly selected cows after their peak of lactation. Milk yield is more stable in that part of the lactation curve, and that also means it's easier to detect any variation among productive data. Several papers in the literature deal with the problem of lag response of daily milk yield to heat stress, thus it's possible to consider it when analysing the data. Moreover, 40 cows per almost 80 days of experimental trial means 3,200 data about daily milk yield (perhaps data are even more since cow are milked 3-time a day). 

Author Response

Please find attached a document with detailed answers to your valuable comments
